

# RBI: a novel algorithm for regulatory-metabolic network model in designing the optimal mutant strain

Ridho Ananda[1,2], Kauthar Mohd Daud[1] and Suhaila Zainudin[1]

[1] Faculty of Information Science & Technology, Universiti Kebangsaan Malaysia, Kajang, Selangor, Malaysia
[2] Industrial Engineering, Telkom University, Purwokerto, Jawa Tengah, Indonesia

Corresponding author
Ridho Ananda,
p122956@siswa.ukm.edu.my

## ABSTRACT

Over the last 20 years, researchers have proposed regulatory-metabolic network models to integrate gene regulatory networks (GRNs) and metabolic networks in *in silico* metabolic engineering, aiming to enhance the production rate of desired metabolites. However, the proposed models are unable to comprehensively include the Boolean rules in the empirical gene regulatory networks (GRNs) and gene-protein-reaction (GPR) interactions. Thus, the types of gene interactions, such as inhibition and activation, are disregarded from the analysis. This may result in sub-optimal model performance. Hence, this article presented a novel model using reliability theory to include Boolean rules in empirical GRNs and GPR rules in the integrating process. The proposed algorithm of this model is termed as a reliability-based integrating (RBI) algorithm. The suggested algorithm had three variants: RBI-T1, RBI-T2, and RBI-T3. The performance of the RBI algorithms was assessed by comparing them with the existing algorithms, using empirical results and validated transcription factors (TF) knockout schemes, and their complexity time was identified. Also, the RBI method was implemented in the design of optimal mutant strains of *Escherichia coli* and *Saccharomyces cerevisiae*. The simulation results indicated that the effectiveness and efficiency of the RBI algorithms are adequately strong and competitive relative to the existing algorithms. Furthermore, the RBI algorithm effectively identified eight schemes capable of enhancing succinate and ethanol production rates by maintaining the survival of microbial strains. Those results demonstrated that the RBI algorithms are recommended for the construction of optimum mutant strains in *in silico* metabolic engineering.

## INTRODUCTION

Cell factories of microbes naturally produce valuable chemicals, including ethanol, succinate, and 2,3-butanediol (*Singhania et al., 2022*). However, the yield and productivity of these metabolites frequently fall short of the theoretical maximum, thereby constraining their industrial applicability. In order to overcome this problem, comprehensive research has focused on genetically engineering microorganisms to improve their metabolic networks, thereby enhancing the efficiency and production rate of desired metabolites.

Here, the strains of *Saccharomyces cerevisiae* and *Escherichia coli* are frequently utilized to be engineered due to the abundance of their genome information, non-pathogenic, and ease of genetic manipulation (*Sharma & Saharan, 2018*). The aim of this engineering is to enable the creation of optimized mutant strains that can produce industrially significant metabolites at economically feasible levels.

Experimental techniques in the wet lab have been extensively utilized to improve the yield of target metabolites, illustrating their efficacy in overproduction. Nonetheless, these techniques are frequently linked to high costs, labor-intensive, and increased error rates, rendering them inefficient for large-scale applications. To address these limitations, *in silico* metabolic engineering approaches have been developed, employing computational modeling to inform and improve experimental workflows in the laboratory (*Joshi & Mishra, 2022*; *Madden et al., 2020*). Among the computational techniques proposed, flux balance analysis (FBA) has become a popular algorithm for predicting and optimizing metabolic networks (*Daud et al., 2023*). FBA enables the estimation of steady-state growth rates and production capacities of desired metabolites through an understanding of metabolic networks, offering a systematic approach to designing the optimal mutant strains.

The FBA algorithm is a computer modeling framework utilized for conducting *in silico* experiments aimed at optimizing metabolite overproduction from mutant strains (*Fell & Small, 1986*). This technique utilizes a linear programming method to identify optimal flux reactions based on metabolic pathways present in genome-scale metabolic network models (GSMMs). The predominant objective function utilized in FBA is biomass synthesis (*Saa et al., 2022*). Besides biomass, FBA can also be used to predict the overproduction of succinic acid, 2,3-butanediol, or ethanol (*Daud et al., 2019*; *Sen, 2022*; *Ananda, Daud & Zainudin, 2023*). FBA has been effectively utilized in various domains, including drug target identification (*Arif & Malim, 2013*; *Mahayuddin & Saifuddin Saif, 2020*; *Mahiddin, Ali Othman & Abdul Rahim, 2021*; *Marin-Hernandez & Saavedra, 2023*) and elucidating cellular responses to varying conditions (*Nurjana & Zainudin, 2018*; *Yasemi & Jolicoeur, 2021*; *Shih & Morgan, 2020*). Notwithstanding its benefits, FBA is unable to integrate gene regulation into the metabolic network when determining the best flux response (*Pereira & Rocha, 2022*). Gene regulation in gene regulatory networks (GRNs) encompasses interactions among genes, including inhibition, repression, and activation, wherein the expression of a gene is modulated by other genes, namely transcription factors (TFs). The condition of the gene, as determined by gene-protein-reaction (GPR) rules, will influence the state of the flux reactions, determining their activity or inactivity (*Chung et al., 2021*). The shortcoming of FBA to incorporate gene regulation will undoubtedly yield inaccurate predictive outcomes. To address this issue, researchers have already proposed computational models that integrate gene regulatory networks and metabolic networks, termed the regulatory-metabolic network model (*Ananda, Daud & Zainudin, 2024*). These models offer extensive insights into the metabolic processes and regulatory mechanisms operating within cells simultaneously. Figure S1 illustrates the primary stages of regulatory-metabolic network models for deriving the optimal mutant strain *via* genetic engineering explained by *Malcı et al. (2023)*. Then, the optimal mutant strains obtained

were utilized for wet lab experiments, which encompass screening, production on a shake flask scale, and production on a bioreactor scale until the overproduction of the desired compounds on an industrial scale is obtained.

The current regulatory-metabolic network models can be categorized into two types: discrete models and continuous models. In discrete models, the states of genes and reaction fluxes, controlled by regulatory constraints, are exclusively categorized as either active or inactive. The algorithms in the discrete category include regulatory flux balance analysis (rFBA) (*Covert & Palsson, 2002*), steady-state regulated flux balance analysis (SR-FBA) (*Shlomi et al., 2007*), integrated flux balance analysis (iFBA) (*Covert et al., 2008*), and the toolbox for integrating genome-scale metabolism, expression, and regulation (TIGER) (*Jensen, Lutz & Papin, 2011*). Although these algorithms have successfully integrated gene regulatory networks with metabolic networks, the rigorous regulation of the regulatory-metabolic networks model may result in erroneous predictions. This happens because, in fact, several genes or transcription factors have just a partial effect on flux processes rather than completely activating or inhibiting them (*Raman, 2021*). To overcome it, *Lee et al. (2007)* modified the rFBA algorithm by involving weights. After that, researchers proposed several algorithms for modeling regulatory-metabolic networks in continuous model, including probabilistic regulation of metabolism (PROM) (*Chandrasekaran & Price, 2010*; *Ma et al., 2015*), transcriptional regulated flux balance analysis (TRFBA) (*Motamedian et al., 2017*), optimization of regulatory and metabolic networks (OptRAM) (*Shen et al., 2019*), and transcription regulation integrated with metabolic regulation (TRIMER) (*Niu et al., 2021*). These algorithms have accurately simulated the partial impacts seen in genes or flux processes when the collection of transcription factors is either activated or eliminated. The PROM and TRFBA algorithms use empirical gene regulatory networks (GRNs) in integrating GRNs and metabolic networks, while OptRAM and TRIMER use the inferred GRNs. The empirical GRNs that represent gene regulation *via* Boolean equations are obtained from direct observations using RNA-seq technologies (*Van Der Sande, Frölich & Van Heeringen, 2023*). Meanwhile, the deduced GRNs are constructed using gene expression data *via* data-driven methodologies, including machine learning approaches (*Zhao et al., 2021*).

Although the continuous model has successfully built non-rigid regulatory constraints, further modifications are necessary. The PROM method solely employs the relative impact of each TF on the regulated gene. Meanwhile, although the TRFBA and OptRAM algorithms effectively incorporate all TFs regulating a gene, they do not account for the interactions, such as inhibition and activation, shown in Boolean rules at the empirical GRNs. These models exclusively use a set of transcription factors that regulate a gene to generate a particular equation by utilizing gene expression data. On the other hand, TRIMER excels at modeling soft regulatory constraints compared to PROM; specifically, it may incorporate many metabolic genes concurrently to establish the limits for the flux reaction in accordance with GPR rules. Nonetheless, similar to TRFBA and OptRAM, TRIMER fails to model the interaction types within GRNs. Those limitations certainly have the potential to produce inaccurate metabolic engineering results.

Based on the statements mentioned above, this article proposes a novel regulatory-metabolic network model known as the reliability-based integrating (RBI) algorithm. The procedure of reliability theory is utilized in RBI to model all TFs and genes influencing the flux reaction comprehensively and simultaneously, taking into account the types of interactions, including inhibition and activation. The RBI algorithm uses the empirical GRNs as they convey information regarding the interaction types between TFs and genes. In addition, the empirical GRNs exhibit greater confidence compared to the inferred GRNs, as they are derived comprehensively from direct experimental data (*Larsen et al., 2019*; *Parise et al., 2021*). By implementing reliability theory, the probabilities of gene states and reaction fluxes can be modeled comprehensively by incorporating the interaction types of TFs and genes involved in the catalysis of reaction fluxes.

Several validations are required to identify the performance of RBI in designing the optimal mutant strains. This study assessed the accuracy of RBI prediction results in specific mutant strains, grounded in empirical experiments and the previous literature, demonstrating the capability to generate mutant strains that enhance the production rate of desired metabolites. The performance was compared to existing algorithms for continuous types, utilizing both the empirical GRNs and the inferred GRNs. Finally, the time complexity of the RBI algorithm was determined.

The primary contributions of this study are enumerated below.

1. A novel algorithm, termed the RBI algorithm, was proposed for integrating GRNs and metabolic networks. This algorithm was incorporated into the continuous model utilizing the empirical GRNs.
2. The performance of the RBI algorithm was evaluated against state-of-the-art algorithms using actual results and previous literature. In addition, the time complexity of the RBI algorithm has been identified.
3. Lastly, the RBI algorithm was implemented in designing the optimal mutant strains of *E. coli* and *S. cerevisiae* to enhance the production rate of the succinate and ethanol.

This article has several parts. The first section provides a brief introduction to reliability theory. The second part details the materials and methods used in this study. Subsequently, the results and discussion are presented in the third and fourth sections. The conclusion is presented in the last section.

## A BRIEF OF RELIABILITY THEORY

Reliability theory is a mathematical discipline that investigates the probability of a system's reliability in relation to its components (*Ross, 2023*). This theory has recently been applied to address challenges in various domains, including waste management systems (*Khristodulo et al., 2020*), safety applications (*Xu & Saleh, 2021*), and manufacturing systems (*Singla, 2023*). The implementations illustrate that reliability theory can effectively address real-world problems.

In order to determine the reliability value of a system, it is necessary to obtain information regarding the probability values of its components and their structural

configurations. Let $\mathbf{x} = (x_1, x_2, \ldots, x_n)$ represent the component vector that affects the reliability of a system ($\phi_{\mathbf{x}}$). The probability value of the $i$th component, denoted as $p_i$, indicates the likelihood of this component being active. There are three types of component structures: series, parallel, and hybrid, where their illustration have been presented in Fig. S2. The formula used to calculate $r_{\phi_{\mathbf{x}}}$ in series and parallel structures is presented in Eq. (1). The reliability value of a system with a hybrid structure can be determined by integrating the series and parallel formulas.

$$r_{\text{series}} = \prod_{i=1}^{n} p_i \qquad \text{and} \qquad r_{\text{Parallel}} = 1 - \prod_{i=1}^{n} (1 - p_i). \qquad (1)$$

This discussion provides a concise technical overview of reliability theory as it pertains to assessing the reliability value of a system, with a comprehensive exposition available in *Ross (2023)*. This study will utilize reliability theory to analyze metabolic genes and reactions as a system. Meanwhile, the regulatory genes or TFs are considered components of systems.

## MATERIALS AND METHODS

### The implementation of reliability theory in metabolic engineering

The RBI algorithm operated reliability theory to determine the reliability value of metabolic genes or reaction fluxes. To calculate the reliability value, the Boolean rules outlined in the empirical GRNs and GPRs must be considered. The Boolean rules governing the empirical GRNs consist of three operators: NOT, AND, and OR. The NOT operator indicates that transcription factors (TFs) associated with this operator inhibit the regulated genes. Suppose that the $i$th transcription factor and the $j$th metabolic gene are denoted by $TF_i$ and $MG_j$, respectively. Several rules must be considered when determining the reliability value of $MG_j$ ($r_{MG_j}$) regulated by $TF_i$, as outlined in the following numbering.

1. If $MG_j$ is only regulated by $TF_i$ involving the NOT operator, *i.e.*, $MG_j$ = NOT $TF_i$, then the reliability value of $MG_j$ is determined by Eq. (2).

$$r_{MG_j} = p_{TF_i}^c \qquad (2)$$

where $p_{TF_i}^c$ is the complement of $p_{TF_i}$, which is the probability value of $TF_i$ being active. If $TF_i$ is knocked out, then $p_{TF_i} = 0$.

2. If $MG_j$ is regulated by $TF_h$ and $TF_i$ involving the AND operator, *i.e.*, $MG_j = TF_h$ AND $TF_i$, then based on the formula of series structure in reliability theory, the reliability value of $MG_j$ is determined by Eq. (3).

$$r_{MG_j} = p_{TF_h} \cdot p_{TF_i}. \qquad (3)$$

The series structure is used in this case due to the same characteristic between the AND operator and this structure.

3. If $MG_j$ is regulated by $TF_h$ and $TF_i$ involving the OR operator, *i.e.*, $MG_j = TF_h$ OR $TF_i$, then based on the formula of parallel structure in reliability theory, the reliability value of $MG_j$ is determined by Eq. (4).

$$r_{MG_j} = 1 - (1 - p_{TF_h})(1 - p_{TF_i}). \qquad (4)$$

The use of parallel structure in this case is also based on the same reasons as the previous operator.

For instance, *aspA*, one of the metabolic genes of *E. coli* type strain iAF1260 is regulated by two TFs, *Crp* and *FnR*, whose Boolean rules are shown in Eq. (5).

$$aspA = (Crp \text{ AND } (\text{NOT Fnr})) \text{ OR Fnr}. \tag{5}$$

There are three operators involved in regulating *aspA*, namely NOT, AND, and OR. The reliability value of *aspA* ($r_{aspA}$) can be determined using the three previously established reliability calculation rules. The illustration of transcription factors regulating the *aspA* gene in a series-parallel structure is in Fig. S3. The steps to get the $r_{aspA}$ formula are elaborated in Eq. (6).

$$
\begin{aligned}
r_{aspA} &= (Crp \text{ AND } (\text{NOT Fnr})) \text{ OR } Fnr \\
&= (p_{Crp} \text{ AND } p_{Fnr}^c) \text{ OR } p_{Fnr} \\
&= 1 - (1 - p_{Crp} \cdot p_{Fnr}^c)(1 - p_{Fnr}).
\end{aligned}
\tag{6}
$$

The same procedure is utilized to determine the reliability value of flux reactions catalyzed by metabolic genes outlined in the GPR rules on GSMMs. The ATP synthase reaction, *ATPS4r*, is a flux reaction in strain iAF1260, catalyzed by three metabolic genes: *AtpF0*, *AtpF1*, and *AtpI*, as indicated by the boolean rule in Eq. (7).

$$ATPS4r = (AtpF0 \text{ AND } AtpF1) \text{ OR } (AtpF0 \text{ AND } AtpF1 \text{ AND } AtpI). \tag{7}$$

The illustration of the metabolic gene structure that catalyses the reaction flux *ATPS4r* has been presented in the Fig. S4. The method for acquiring $r_{ATPS4r}$ is outlined in Eq. (8).

$$
\begin{aligned}
r_{ATPS4r} &= (AtpF0 \text{ AND } AtpF1) \text{ OR } (AtpF0 \text{ AND } AtpF1 \text{ AND } AtpI) \\
&= (p_{AtpF0} \text{ AND } p_{AtpF1}) \text{ OR } (p_{AtpF0} \text{ AND } p_{AtpF1} \text{ AND } p_{AtpI}) \\
&= (p_{AtpF0} \cdot p_{AtpF1}) \text{ OR } (p_{AtpF0} \cdot p_{AtpF1} \cdot p_{AtpI}) \\
&= 1 - (1 - p_{AtpF0} \cdot p_{AtpF1})(1 - p_{AtpF0} \cdot p_{AtpF1} \cdot p_{AtpI}).
\end{aligned}
\tag{8}
$$

The calculations presented will be utilized for all genes regulated by TFs in GRNs to determine their reliability values. Also, the reliability value of reaction flux based on the GPR rules is calculated using the same procedure. Subsequently, those reliability values obtained will be used to establish soft constraints for the regulatory-metabolic networks model, aiming to identify optimal mutant strains.

## The reliability-based integrating algorithm

This study presents a novel algorithm for constructing a regulatory-metabolic network model. This approach aims to establish a comprehensive pipeline linking GRNs and metabolic networks that incorporates the interaction types between various genes in GRNs and GPRs, a gap not addressed by existing algorithms. The algorithm proposed is the reliability-based integrating (RBI) algorithm. The reliability theory is applied in RBI to address the Boolean rules found in the empirical GRNs and GPR rules.

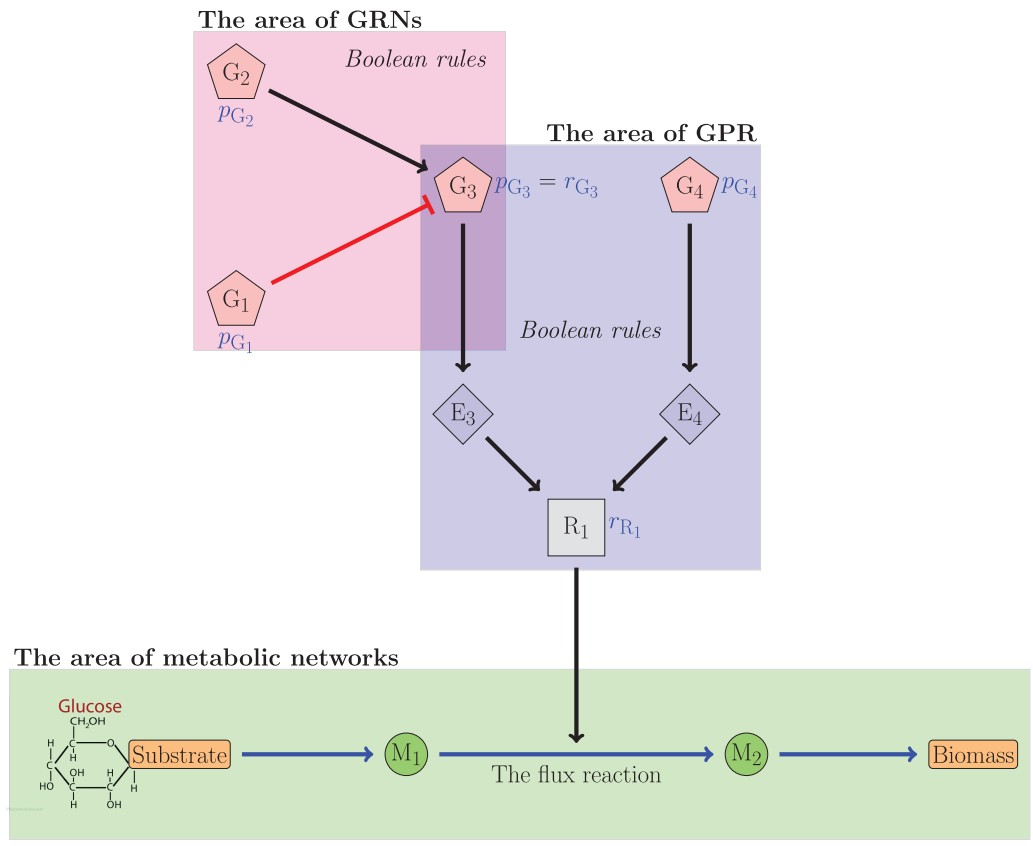

**Figure 1 The relationship illustration of GRNs and the metabolic networks.**

Figure 1 illustrates the relationship between gene regulatory networks and metabolic networks. In the figure, the $i$th gene was denoted by $G_i, \forall i \in \{1, 2, 3, 4\}$. The $j$th enzyme was denoted by $E_j, \forall j \in \{3, 4\}$. The $k$ metabolite was denoted by $M_k, \forall k \in \{1, 2\}$. Meanwhile, $R_1$ was the reliability value of the flux reaction catalyzed by $G_3$ and $G_4$. The figure indicated that $G_3$ was regulated by $G_1$ and $G_2$, which respectively inhibited and activated $G_3$. The reliability value of $G_3$ can be determined from this interaction. The reliability value of $R_1$ was subsequently calculated using the Boolean equation involving $G_3$ and $G_4$. The reliability value of the flux reactions will be used to establish regulatory constraints governing the flux reaction. In this case, the reliability theory is utilized to compute $r_{G_3}$ and $r_{R_1}$.

Three types of RBI algorithms have been proposed in this study, namely RBI-T1, RBI-T2, and RBI-T3. The distinction is in establishing the initial probability value of TFs. The RBI-T1 algorithm assumes that TFs that are not knocked out have the same probability of being in active or inactive conditions. Consequently, the initial probability value of TFs in this algorithm is set to 0.5. The initial probability values of RBI-T2 and RBI-T3 are based on gene expression data. The RBI-T2 algorithm utilizes the binarization process from the PROM algorithm to derive the probability of active

transcription factors (TFs). The initial probability value of RBI-T3 is derived from the average of the initial probability values of RBI-T1 and RBI-T2. The formula to get the initial probability values of transcription factors for each type of RBI algorithm is presented in Table S1.

Furthermore, the RBI algorithm considers three parameters: $\alpha$, $\beta$, and $\gamma$. The $\alpha$ parameter is derived from the binarization of gene expression data to establish the initial values of transcription factors for RBI-T2 and RBI-T3. This parameter is used to acquire information regarding the probability value of genes as transcription factors in an active state, utilizing the abundance of gene expression data measured under diverse environmental conditions and genetic perturbations (*Chandrasekaran & Price, 2010*). Additionally, the $\beta$ and $\gamma$ parameters are utilized by RBI-T1, RBI-T2, and RBI-T3 to model the partial effects experienced by various reactions as a result of the influence of genes or transcription factors (*Raman, 2021*; *Simeonidis, Chandrasekaran & Price, 2013*). The block diagram of the RBI procedure is available in Fig. S5. The technical steps of this algorithm are outlined below.

1. Input data required, namely GSMMs, GRNs, and the gene expression data.

2. Choose the TFs to be knocked out.

3. Determine the initial probability value of $\text{TF}_i, \forall i$ using Eq. (9).

$$p_{\text{TF}_i} = \begin{cases} 0 & \text{if the } i\text{th TF is knocked out} \\ f(\text{TF}_i) & \text{otherwise.} \end{cases} \tag{9}$$

Determining the value of $f(\text{TF}_i)$ is based on the RBI type. For RBI-T1, $f(\text{TF}_i) = 0.5, \forall i$. Furthermore, if RBI-T2 is used, the value of $f(\text{TF}_i)$ is obtained from Eq. (10).

$$f(\text{TF}_i) = \frac{1}{d} \times n(\text{TF}_i = 1). \tag{10}$$

where $n(\text{TF}_i = 1)$ refers to the number of '1' values in the gene expression data that has been transformed to binary form based on the parameter $\alpha$ and the binarization procedure of the PROM algorithm (*Chandrasekaran & Price, 2010*; *Ananda, Daud & Zainudin, 2024*); meanwhile, $d$ is the dimension of the gene expression data used. Moreover, $f(\text{TF}_i)$ of RBI-T3 is obtained from the average value of $f(\text{TF}_i)$ obtained by RBI-T1 and RBI-T2, the formula is presented in Eq. (11).

$$f(\text{TF}_i) = \frac{1}{2}\left(0.5 + \frac{1}{d} \times n(\text{TF}_i = 1)\right). \tag{11}$$

4. Calculate $r_{\text{MG}_j}, \forall j$, namely the reliability value of the $j$th metabolic gene, using the reliability theory formula.

5. Define the probability value of the metabolic genes obtained from their reliability values when those genes are regulated by their respective TFs. However, if there is no information that TFs regulate metabolic genes, their probability is one.

6. Calculate $r_{\text{RF}_k}, \forall k$, namely the reliability value of the $k$th reaction flux, using the reliability theory formula.

7. Determine the reference flux using flux variability analysts (FVA). We obtain $[lb_k^*, ub_k^*]$, $\forall k$ which is the bounds of the $k$th reaction flux.

8. Generate regulatory constraints into the model by updating the bounds of reaction fluxes (RF) that are not transport reactions or 'ATPM' reactions and whose reliability values are less than or equal to $\beta$. The formula used to update the bounds of the $k$th reaction flux, $\forall k$, is shown in Eq. (12).

$$ub_k = r_{RF_k} \times ub_k^* + \gamma.$$
$$lb_k = r_{RF_k} \times lb_k^* - \gamma.$$
(12)

where $\gamma$ refers to the soft constraint adopted from the PROM algorithm (*Chandrasekaran & Price, 2010*).

9. Determine the optimal reaction flux using the FBA algorithm.

10. The optimal flux distribution is obtained.

## The dataset used

To validate the performance of the proposed RBI algorithms, a simulation of metabolic engineering was performed using those algorithms on *E. coli* and *S. cerevisiae*. The results were compared with empirical data and previous literature demonstrating the potential mutant strains for the overproduction of desired metabolites. The computational simulation conducted utilized genome-scale metabolic models (GSMMs), regulatory networks (GRNs), and gene expression data as datasets.

### Genome-scale metabolic model

GSMMs represent comprehensive frameworks of metabolic networks, integrating metabolic data, including flux reactions and their directionalities, associated catalyzing enzymes, and GPR associations. This study utilized the GSMMs iAF1260 and Yeast 7.6, representing strain models of *E. coli* and *S. cerevisiae*, respectively. The iAF1260 model was utilized to estimate the biomass and indole production rates (*Niu et al., 2021*). Meanwhile, the Yeast 7.6 model was used to forecast the flux range of succinate, 2,3-butanediol, and ethanol (*Shen et al., 2019*; *Man et al., 2021*; *Cruz et al., 2024*; *Shirai & Kondo, 2022*). Additional details regarding those strains are provided in Table S2.

### Gene regulatory networks

The datasets for gene regulatory networks utilized were iMC1010 and iMH805/837, representing gene regulation models for the microorganisms *E. coli* and *S. cerevisiae*, respectively. The data can be accessed at https://systemsbiology.ucsd.edu/. The data consists of the empirical GRNs that encompass the interactions of TFs represented in Boolean equations. The iMC1010 dataset comprises 104 TFs and 479 metabolic genes that their TFs regulate. This data has been utilized to reconstruct the integration model of gene regulation and metabolism in *E. coli* (*Grimbs et al., 2019*; *Niu et al., 2022*). GRNs iMH805/837 comprise 55 transcription factors and 837 regulatory interactions derived from the primary literature (*Herrgård et al., 2006*).

### The gene expression data

Furthermore, the gene expression data from microbial *E. coli*, comprising 213 normalized Affymetrix microarray profiles derived from various published studies conducted in the Palsson Lab, as summarised by *Lewis et al. (2009)*, was utilized in this study. The data can be accessed at https://systemsbiology.ucsd.edu/. Meanwhile, the gene expression data for *S. cerevisiae* were sourced from the Gene Expression Omnibus (GEO) repository. The data were produced by *Sardi et al. (2018)* in their investigation of the response of *S. cerevisiae* to 1-butanol, isobutanol, or ethanol across various genetic backgrounds and tolerances under both aerobic and anaerobic conditions.

## The performance measurements

Performance measurements are essential for assessing the efficacy of the proposed RBI algorithms. The simulations assessed both effectiveness and efficiency. The validation applied to assess these two aspects is outlined in the subsequent discussion.

### The validity measures

Validity measures denote the techniques employed to assess the performance of the algorithm. The validity pertains to the precision of the prediction in relation to the actual outcome. This study used several validity measures, including the root mean square error (RMSE) (*Arif, Fatimah Zawani Abdullah & Malim, 2014*; *Gonçalves, Henriques & Costa, 2023*), the Pearson correlation coefficient (PCC) ($\rho$) (*Niu et al., 2021*; *Caivano et al., 2023*), R-squared ($R^2$) (*Yadav et al., 2023*; *Zhang et al., 2020*), and the bias value (*Lee & Nam, 2022*; *Tsimenidis, Vrochidou & Papakostas, 2022*; *Odrzywolek et al., 2022*). The formula for these measurements is presented in Eqs. (13)–(16).

$$\text{RMSE} = \sqrt{\frac{1}{n} \sum_{i=1}^{n} (y_i - \hat{y}_i)^2}. \tag{13}$$

$$\rho = \frac{\text{cov}(\mathbf{y}, \hat{\mathbf{y}})}{\sigma_{\mathbf{y}} \sigma_{\hat{\mathbf{y}}}}. \tag{14}$$

$$R^2 = 1 - \frac{\text{SSE}}{\text{SST}}. \tag{15}$$

$$\text{The Bias value} = \frac{\sum_{i=1}^{n} (\hat{y}_i - y_i)}{n} \tag{16}$$

where $y_i \in \mathbf{y}$ and $\hat{y}_i \in \hat{\mathbf{y}}, \forall i$ are the actual results and the predicted values, respectively. The value of $n$ refers to the number of observations. Additionally, 'cov' and $\sigma$ are the covariance and the standard deviation, respectively. Furthermore, SSE and SST are the unexplained variation and the total variation, respectively.

The RMSE value indicates the accuracy of the algorithm used, where the lower the RMSE, the higher the accuracy. Meanwhile, PCC ($\rho$) shows the value of a linear correlation between the prediction and actual results. The PCC value close to 1 means that a high positive correlation is obtained. Conversely, this value was close to −1, indicating a high negative correlation. Meanwhile, PCC approximates to zeros, implying there is lacking linear correlation. At the same time, R-square ($R^2$) notifies how suitable the prediction

results of particular algorithms are to the actual results in explaining its variance. Finally, the bias value describes the systematic error within the prediction results. A positive bias denotes the overestimated prediction obtained. Conversely, a negative bias points to the underestimated prediction. The good performance of an algorithm based on the bias value is obtained when its bias value is almost zero.

### Time complexity

Time complexity was utilized to validate the efficiency of the RBI algorithm during *in silico* experiments. Its time complexity was diagnosed based on big O analysis within each part of RBI. This procedure was provided by *Odhiambo Omuya, Onyango Okeyo & Waema Kimwele (2021)*.

## RESULTS

### The optimal parameters of the RBI algorithm

The RBI algorithms necessitate three parameters ($\alpha$, $\beta$, and $\gamma$), which emerge from uncertainties in genes or reaction fluxes. Variability in gene expression significantly impacts cellular function. It arises from various factors, including stress response, pathogenesis, metabolism, development, cell cycle, circadian rhythms, and ageing (*Raj & Van Oudenaarden, 2008*). RBI-T2 and RBI-T3 use the $\alpha$ parameter to model uncertainties and derive initial quantity measures of regulatory gene expression or TFs from gene expression data, utilizing the binarization procedure of the PROM algorithm (*Chandrasekaran & Price, 2010*). The values serve as initial probability values for transcription factors in the RBI algorithm. Then, the $\beta$ and $\gamma$ parameters are utilized due to the reality that numerous genes/transcription factors may exert only a partial influence on reaction fluxes (*Raman, 2021*). The parameter $\beta$ quantifies the extent of enzyme performance generated by metabolic genes in affecting the activation energy of a flux reaction (*Urry et al., 2017*). Meanwhile, $\gamma$ serves as a partial effect for the bounds of flux reactions.

To obtain the optimal parameters required by RBI, various metaheuristic optimization techniques were operated, such as particle swarm optimization (PSO), genetic algorithm (GA), simulated annealing (SA), differential search algorithm (DSA), bijective-DSA (B-DSA), surjective-DSA (S-DSA), Elitist(1)-DSA (E1-DSA), Elitist(2)-DSA (E2-DSA), grey wolf optimization (GWO), whale optimization algorithm (WOA), genetic algorithm based on natural selection (GABONST), komodo mlipir algorithm (KMA), and Aquila optimizer (AO). Those selected metaheuristic optimisations were either state-of-the-art metaheuristic optimisation techniques or had been utilized in the design of the optimal mutant strain (*Hor et al., 2024*; *Tan et al., 2023*; *Dzulkalnine et al., 2023*; *Suyanto, Ariyanto & Ariyanto, 2022*; *Abualigah et al., 2021*; *Lee et al., 2020*; *Albadr et al., 2020*; *Shen et al., 2019*; *Mirjalili & Lewis, 2016*; *Mirjalili, Mirjalili & Lewis, 2014*). The fitness value was derived from the multiplicative inverses of the RMSE obtained.

Two microbial strains, *E. coli* and *S. cerevisiae*, have been used in simulations to determine the optimal parameters for each type of RBI. The results and gene knockout schemes for *E. coli* were presented by *Covert et al. (2004)*, with biomass flux serving as the

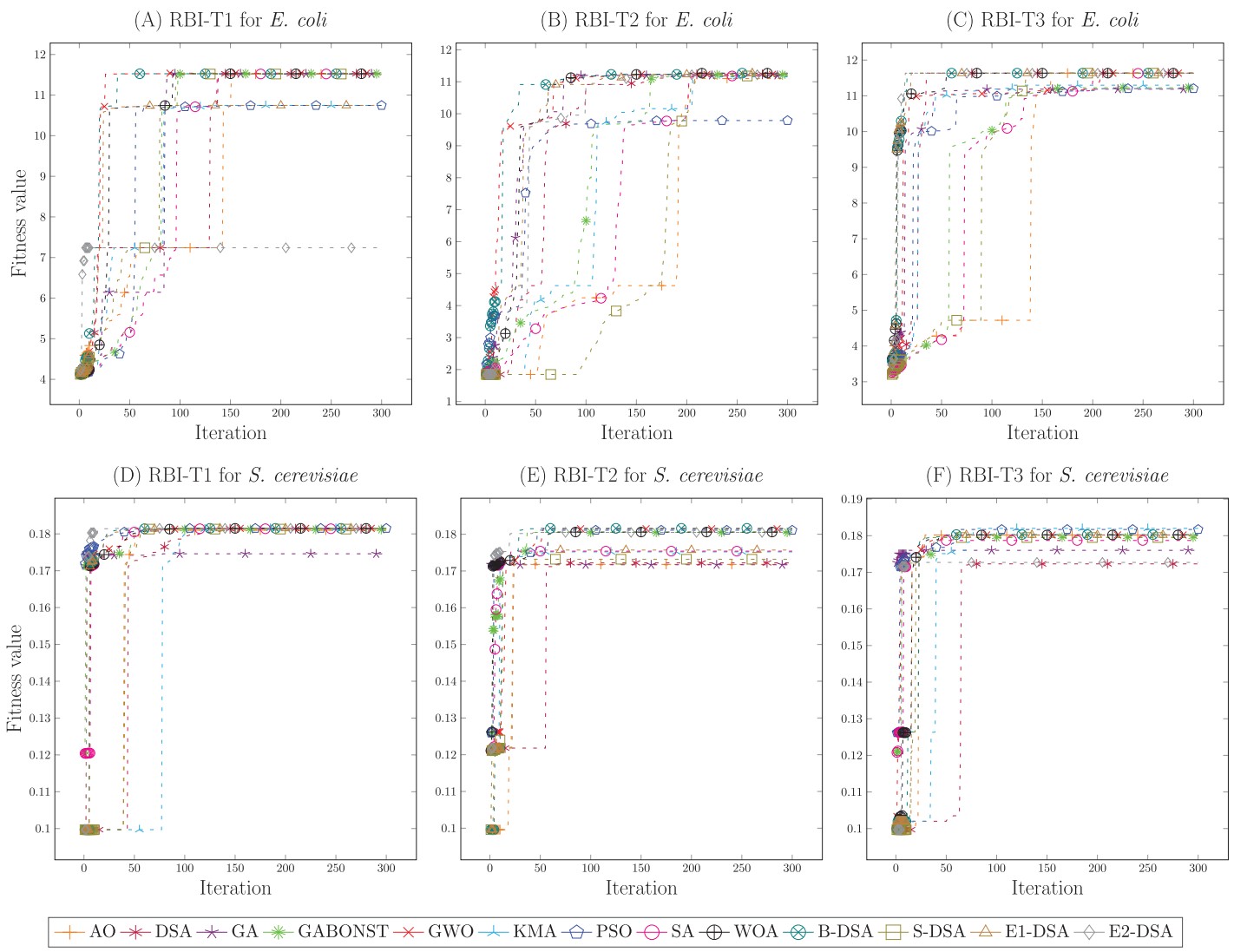

**Figure 2** **The convergence curve of the fitness value of each metaheuristic in determining the optimal parameters of the RBI algorithms.** The simulations of the *E. coli* strain were presented by (A)–(C). Meanwhile, the *S. cerevisiae* strain were given in (D)–(F).

objective function. In *S. cerevisiae*, the production rates of succinate, 2, 3-butanediol, and ethanol were predicted and compared with previous literature (*Shen et al., 2019*). Those empirical results and validated TF knockout scheme were utilized to get the valid parameters for RBI, where to get the valid parameters, the empirical data was required *Helbing (2010)*.

Figures 2A–2C demonstrated the simulation results of the *E. coli* strain each RBI type. From Fig. 2A, it could be known that the global convergence of B-DSA, GWO, S-DSA, DSA, WOA, and GABONST was similar enough. However, GWO precisely obtained the highest global convergence with a small difference; namely, its fitness value was 11.562. Then, in Fig. 2B, WOA received the highest global convergence of 11.269, demonstrating a

small difference to SA, B-DSA, S-DSA, E1-DSA, KMA, GWO, GA, and GABONST. Afterward, the parameters achieved by the DSA algorithm for RBI-T3 were chosen because its algorithm provided the highest global convergence, namely 11.634. Accordingly, the optimal parameters of RBI-T1, RBI-T2, and RBI-T3 were respectively obtained from GWO, WOA, and DSA. Meanwhile, Figs. 2D–2F showed the simulation results of the *S. cerevisiae* strain in determining the optimal parameters of each RBI type. Based on the fitness value compared, it was concluded that the E1-DSA algorithm provided the best optimal parameters, where its fitness value of 0.1815 was the highest value. Furthermore, Figs. 2E and 2F showed that the B-DSA and KMA algorithms acquired the same and best global convergence, namely 0.182. Hence, the best optimal parameters for RBI-T1, RBI-T2, and RBI-T3 in the *S. cerevisiae* strain, respectively were obtained from E1-DSA, B-DSA, and KMA. Furthermore, the optimal parameters used in this study for each RBI type are presented in Table S3.

## Performance of the RBI algorithm

The RBI algorithm was developed to integrate GRNs and metabolic networks by incorporating comprehensive Boolean rules from both GRNs and GPRs, including their interaction types. This algorithm enables researchers to assess the effectiveness of gene engineering approaches and determine the most effective mutant strategy for optimizing the desired metabolic flux. The designed mutant strains can be estimated *in silico* using RBI. The estimation results can subsequently be validated through actual TF knock-out and screening experiments conducted in the wet lab. The RBI algorithms were applied to estimate the outcomes of its gene engineering based on the actual results and prior literature, which were provided by *Niu et al. (2021)* and *Shen et al. (2019)*.

The previous algorithms for the regulatory-metabolic network model were employed to compare and assess the RBI performance, specifically continuous-type models utilizing the empirical GRNs or the inferred GRNs. The advanced algorithms employed in the empirical GRNs were PROM and TRFBA. Simultaneously, recent algorithms utilizing the inferred GRNs, including TRIMER, OptRAM, and OptFlux, were utilized.

### Accuracy in production rate of biomass and indole

We conducted a comparison of the RBI algorithm's prediction results for reaction fluxes with those of other algorithms to demonstrate its predictive capability. We assessed the accuracy of the biomass and indole flux predictions generated by those computational algorithms against actual empirical results. The RBI algorithm was compared with PROM, TRFBA, and TRIMER, with empirical results presented by *Niu et al. (2021)*. In this simulation, all types of RBI algorithms have been simulated. Meanwhile, other algorithms have only utilized their optimal outcomes from previous work.

The first comparison was conducted based on the biomass prediction results using the microorganism *E. coli* strain type iAF1260. Table S4 presented the biomass flux prediction results influenced by perturbations from TF knockout mutants, specifically arcA, fnr, appY, oxyR, and soxS, under aerobic and anaerobic conditions. In this context, we define

WT (wild type) as a type that estimates strain biomass without the presence of the TF knockout mutant. We utilized four validation measures: RMSE, PCC, R-squared, and bias value. These measures provide insights into the accuracy, correlation, and systematic error of each algorithm. Then, the values highlighted in bold indicate the optimal algorithm for the specific validity measure.

The comparative results in Table S4 indicated that RBI-T1 demonstrated good performance across all the used validity measures, achieving the lowest RMSE of 0.088, and the PCC (0.871) and R-square (0.647) values of RBI-T1 were competitive, even if slightly lower than those of TRFBA and TRIMER. Its bias value of 0.059 was relatively small, indicating an overestimation. Based on those results, the predictive capability of RBI-T1 exhibited significant reliability and accuracy in this instance. Meanwhile, the prediction results of RBI-T2 demonstrated sub-optimal quality according to all the validity measures, with RMSE, PCC, R-squared, and bias values of 0.634, −0.737, 0.474, and 0.589, respectively, as presented in Table S4. The results indicated that RBI-T2 showed low accuracy and inconsistent prediction results. Conversely, RBI-T3 demonstrated preferable validity measurement results relative to RBI-T2. However, it was inferior to PROM, TRFBA, and TRIMER. The RMSE value of RBI-T3 was 0.247, indicating moderate performance. The PCC value of 0.296 and R-squared of 0.473 suggested a sufficient positive prediction correlation and adequate capacity to explain the variance in the empirical data. The bias of RBI-T3, measured at 0.187, implied a slight overestimation in the prediction results. RBI-T3 revealed a more satisfactory performance relative to RBI-T2. Furthermore, RBI-T3 demonstrated potential as an alternative algorithm.

Likewise, we conducted a second comparison to evaluate the RBI algorithm's ability to predict the indole production rate. The perturbations employed were some TF knockout mutants from Niu et al. (2021), for which their empirical GRNs were available, namely Fnr, soxS, Crp, lysR, fucR, Mali, phoB, cpxR, tnaA, and tnaB. Similar to the first comparison, the prediction results of the RBI algorithms were compared with the results of PROM, TRFBA, and TRIMER, which used flux balance analysis (FBA) to predict the optimal flux reactions. In this case, the type of TRIMER used was TRIMER-C because its prediction results were best. The validity measures used were RMSE, PCC, R-squared, and bias value. Results of the comparison have been presented in Table S5.

Based on the comparison results shown in Table S5, RBI-T1 got better RMSE (0.025) and bias (0.025) values compared to TRFBA and better R-squared (0.500) values compared to PROM and TRIMER. In addition, the PCC value (0.499) of RBI-T1 showed a positive correlation between the prediction results and the actual empirical results and was moderate. This indicated that RBI-T1 had a competitive performance compared to other algorithms in this simulation. On the other hand, RBI-T2 performed better than RBI-T1 in almost all the validity measures used. The accuracy of RBI-T2, as indicated by the RMSE and bias values (0.023), was better than TRFBA, although lower than PROM and TRIMER. The PCC value of RBI-T2, namely 0.808, outperformed the values of PROM, TRFBA, and TRIMER. The obtained PCC value showed a strong positive correlation between the RBI-T2 prediction results and the actual empirical results, indicating a high goodness of fit

in the prediction results. Likewise, in the R-squared value, RBI-T2 obtained a value of 0.500, which suggested the ability to explain empirical data variance better than PROM, TRFBA, and TRIMER. Those descriptions showed the ability of the RBI-T2 algorithm to predict the production rate of indole for *E. coli* was better than RBI-T1, PROM, TRFBA, and TRIMER in the consistency of prediction results and the ability to explain empirical data variance. Meanwhile, the performance of the RBI-T3 algorithm based on the table showed a lower performance than RBI-T2 and RBI-T3 in this case. The RMSE and bias values of RBI-T3 were relatively low but still above RBI-T1 and RBI-T2. Then, when considered from the R-squared value, the RBI-T3 algorithm showed moderate ability in explaining empirical data variance. However, its PCC of −0.753 showed inconsistency between the prediction results and the actual empirical results, even though the correlation was quite high. This indicates that RBI-T3 still needs improvement, especially in the suitability of its prediction results with empirical data based on the PCC value in this case.

The prediction results of indole production for each TF KO scheme, as shown in Table S5, yielded a sufficiently similar value. This was also seen in the actual results. Based on the identification of the relationship between genes in TF KO and the reactions influenced, those results occurred because the knocked-out genes in each TF KO impact several metabolite reactions with high similarity. For instance, the Fnr and SoxS impact the same metabolic reactions with a similarity of 0.94. This means that almost all metabolic reactions influenced by Fnr are also influenced by SoxS. This also happens among schemes in TF KO. Consequently, the predicted outcomes of indole synthesis across each TF KO strategy exhibit little variation in the production rate of indole. In fact, in the TRFBA algorithm, the same prediction results are obtained in each TF KO scheme, indicating intuitively that this algorithm exhibits less sensitivity to the TF KO used.

### Accuracy in production rate of succinate, 2,3-butanediol, and ethanol

Also, the proposed algorithms were validated for their predictive capability regarding several valuable compounds produced by yeast, especially succinate, 2,3-butanediol, and ethanol. The prediction results were compared with previous literature, where these compounds have been successfully experimentally enhanced in yeast guided by *in silico* metabolic engineering. The succinate reference was derived from *Otero et al. (2013)*, which presented experimentally validated results from mutant yeast demonstrating enhanced succinate production. Meanwhile, references for butanediol and ethanol were sourced from *Ng et al. (2012)* and *Lisha & Sarkar (2014)*, respectively.

This activity involved a comparison of RBI algorithms with PROM, TRFBA, OptFlux, and OptRAM to predict the lower and upper bounds of the production rate of valuable compounds. The optimization algorithm employed to forecast the production rate boundaries was flux variability analysis (FVA). The perturbation schemes used were TF knockout, as derived from *Shen et al. (2019)* and detailed in Table S6. In

addition, the validation measures utilized consist of RMSE, PCC, R-squared, and the bias value.

The results provided in Table S7 presented that RBI-T1 got superior performance relative to other RBI models. The model exhibited a low RMSE of 6.444, even though it was slightly higher than OptFlux. Furthermore, it attained the highest PCC (0.819), indicating a robust positive correlation between the predicted and observed values. The R-squared value of 0.667 signified that 66.7% of the variability in the data was accounted for by the model. RBI-T1 demonstrated the lowest bias (2.853), indicating minimal systematic deviation in its predictions. Meanwhile, RBI-T2 displayed inferior performance compared to RBI-T1 across all validity measures used. The RMSE of 14.365 was higher than RBI-T1, RBI-T3, OptFlux, and OptRAM, indicating a greater prediction error in RBI-T2. Nevertheless, its accuracy still outperformed PROM and TRFBA. The PCC of 0.305 indicated a moderate correlation between predictions and observations, outperforming OptRAM. The R-squared value was 0.520, indicating that it accounted for 52% of the variability in the data, which was higher than PROM, TRFBA, OptFlux, and OptRAM. RBI-T2 exhibited a higher bias of 6.417 than RBI-T1, signifying an increased systematic error. On the other hand, RBI-T3 performed moderately, surpassing RBI-T2 but underperforming RBI-T1. The RMSE was 6.784, exceeding that of RBI-T1 but lower than RBI-T2, PROM, TRFBA, and OptRAM, showing moderate accuracy performance. The PCC of 0.811 indicated a stronger correlation than RBI-T2, PROM, TRFBA, OptFlux, and OptRAM, albeit weaker than RBI-T1. The R-squared value of 0.657 signified that RBI-T3 accounted for 65.7% of the variability in the data. The bias (3.254) is lower than that of RBI-T2, PROM, TRFBA, and OptRAM but higher than RBI-T1 and OptFlux, suggesting a moderate level of systematic error.

## The time complexity

There are three types of the RBI algorithm, namely RBI-T1, RBI-T2, and RBI-T3. The difference between them is only in determining the probability value of the regulatory genes, $f(\mathrm{RG}_i)\forall i$. In computation, this difference does not have a significant effect on their processing time because the most complex type of RBI uses linear functions to determine the probability values of $\mathrm{RG}_i$. The RBI algorithms utilize four pieces of information during its process: metabolic genes, regulatory genes, metabolic reactions, and metabolites. Suppose that the size of metabolic genes, regulatory genes, metabolic reactions, and metabolites in the dataset are $n_1, n_2, n_3$ and $n_4$, respectively. Subsequently, the time complexity of this algorithm based on big-O notation can be identified based on these sizes. The pseudocode of the RBI algorithm and its big-O notation in each step have been provided in Table S8.

Based on the dataset provided in the Table S2, it was known that the number of reactions, metabolites, and genes in the metabolic networks are not significantly different ($n_1 \approx n_3 \approx n_4$). Meanwhile, the number of regulatory genes was quite below the number of metabolic genes, $n_2 \leq n_1$ (*Shlomi et al., 2007*). Suppose that $n$ is more or equal to $\max\{n_1, n_2, n_3, n_4\}$. Subsequently, the time complexity can be estimated by referring to the size $n$, where the time complexity calculation procedure of the RBI algorithm could be

conducted based on (*Odhiambo Omuya, Onyango Okeyo & Waema Kimwele, 2021*). Equation (17) showed the time complexity result of the RBI algorithm.

$$T(n) = 2O(n_1) + 3O(n_3) + 2O(n_4) + O(n_4.n_3)$$
$$\Leftrightarrow T(n) = 2O(n) + 3O(n) + 2O(n) + O(n^2)$$
$$\Leftrightarrow T(n) = 7O(n) + O(n^2)$$
$$\Leftrightarrow T(n) = O(n^2). \tag{17}$$

Based on the analysis conducted, the time complexity of the RBI algorithm is $O(n^2)$ or quadratic, where the step causing the high time complexity is determining the reference fluxes using FVA. This is reasonable enough because, in the FVA algorithm, the optimization process involves all metabolites, which are carried out repeatedly to predict the boundaries of all metabolic reactions.

Subsequently, simulations were conducted to assess the efficiency of the RBI algorithm in large-scale metabolic networks by measuring its running time. This simulation compared RBI-T1, RBI-T2, and RBI-T3 with PROM and TRFBA, which are the regulatory-metabolic network models that used the empirical GRNs. Additionally, five strains were employed: *E. coli* core, iAF1260, iJO1366, iMM904, iTO977, and Yeast7.6. Those five strain models had varying sizes of genetic regulatory networks and metabolic networks based on the number of metabolites, reactions, genes, and transcription factors provided in Table S9. In this simulation, the objective function used was the production rate of biomass.

Figure S6 showed a line graph depicting the running time of the RBI-T1, RBI-T2, RBI-T3, PROM and TRFBA algorithms on a particular model. From the graph, it is known that the RBI-T1, RBI-T2, RBI-T3, and PROM required relatively the same running time for each strain model. In addition, the increase in running time required by those algorithms is not significant. In contrast, the TRFBA algorithm requires a longer running time with a significant increase along with the increasing size of the strain model. The high running time of TRFBA might be due to the tuning process of the regulatory constraints that had a high complexity and the optimization process that used the mixed integer linear programming (MILP) model. Meanwhile, all types of RBI and PROM algorithms use FBA optimization processes that utilize a linear programming approach and do not contain a fairly complex tuning process for regulatory constraints. The average running time values obtained by RBI-T1, RBI-T2, RBI-T3, PROM, and TRFBA were 1.680, 1.869, 2.553, 1.800, and 37.258, respectively. Quantitatively, it indicated that RBI-T1 is the algorithm with the highest level of efficiency in this case, followed by PROM, RBI-T2, RBI-T3, and TRFBA.

## Designing the optimal mutant strains

A comparative analysis of the proposed algorithms (RBI-T1, RBI-T2, and RBI-T3) has been performed using empirical data and validated potential TF Knockout schemes. This comparison demonstrated accurate predictions and a good correlation between the predicted outcomes and the used benchmark results. The findings indicated that the proposed algorithms were both effective and efficient in the integration of GRNs with metabolic networks for. Accordingly, in this section, the RBI-T1, RBI-T2, and RBI-T3

algorithms were utilized to engineer optimum mutant strains in *E. coli* and *S. cerevisiae* microorganisms. The design outcomes of these algorithms were compared with the PROM and TRFBA algorithms due to both algorithms' capacity to include the empirical GRNs. The prediction results of the wild type were also provided to demonstrate the potential of the acquired mutants.

The strain models used to develop optimum mutant strains, as seen in Table S9, include *E. coli* and *S. cerevisiae* models. The *E. coli* model has been extensively used in metabolic engineering to predict growth rates and the production rates of targeted metabolites, such as succinate. The *E. coli* core and iAF1260 models have been used to assess succinate production rates using flux balance analysis (FBA) combined with a metaheuristic algorithm (*Daud et al., 2019*). Then, iJO1366 has been analyzed using the MoMA algorithm to forecast succinate production rates (*Mienda, Shamsir & Illias, 2016*). Likewise, the *S. cerevisiae* model has been extensively researched owing to the extensive understanding of its metabolism (*Pereira, Nielsen & Rocha, 2016*). Models of *S. cerevisiae*, including iTO977, iMM904, and Yeast 7.0, have been used to forecast significant compounds such as ethanol (*Lopes & Rocha, 2017*). Moreover, the yeast *S. cerevisiae* is the most extensively researched yeast and is used in industrial applications, including ethanol production.

Table S10 provided the predictive outcomes of the succinate production from each algorithm under both aerobic and anaerobic conditions for each strain. The substrates used for the simulation were glucose and oxygen, with the substrate uptake rate following *Niu et al. (2021)* in predicting the growth rate of *E. coli*. The proposed algorithms demonstrated excellent succinate production relative to PROM and TRFBA in almost all strains used, except in *E. coli core*, where its performance was inferior to them. The *E. coli* core model represents the most basic representation of the *E. coli* strain. Therefore, despite the proposed algorithms obtained a lower prediction in *E. coli* core, they effectively optimize the information in the advanced *E. coli* models, achieving a significantly higher production rate in comparison to PROM and TRFBA. Hence, the proposed algorithm generally outperformed PROM and TRFBA in succinate prediction. In addition, those results indicated the potential overproduction of the succinate production due to more than the prediction results of the wild type.

Meanwhile, in the results of designing the optimal mutant strain of *S. cerevisiae* for predicting ethanol production, the suggested algorithms provided competitive results compared to PROM and TRFBA, as seen in Table S11. The predictive outcomes of the RBI-T1 algorithm were marginally worse than those of PROM and TRFBA for strains iMM904 and Yeast7.6. Nonetheless, the predictive outcomes for strain iTO977 were superior. Conversely, the RBI-T2 algorithm demonstrated superior predictive outcomes compared to PROM and TRFBA for strains iTO977 and Yeast 7.6 under anaerobic circumstances, with Yeast 7.6 exhibiting the maximum ethanol output. At the same time, the RBI-T3 exhibited performance almost similar to that of the RBI-T1, which surpasses the iTO977 strain, where the forecast of ethanol production from this strain is the highest. On the other hand, based on the wild-type results, the proposed algorithm showed the potential overproduction in the particular strains, including iTO977 and Yeast 7.6, but

they are under wild-type in iMM904. Nonetheless, the difference in the prediction results at iMM904 was not great, about 0.440 mmol/gDW/hr.

Subsequently, Table S12 presented the results of the optimal mutant strain schemes derived from the proposed algorithm. The findings were chosen based on the maximum metabolite production attained by the RBI algorithms, as seen in Table S10 and S11. Also, the growth rate corresponding to the production rate of the predicted metabolite was provided to identify the optimal design of the growth-coupled (GC) mutant strain capable of surviving post-mutation. The methodology used to acquire the GC mutant strain followed that of *Daud et al. (2019)*, while the substrate uptake was based on the simulation performed by *Niu et al. (2021)*. The table presented a scheme of mutant strains that might enhance succinate synthesis, compared to the wild-type, in the strains *E. coli* core (ANA), iAF1260 (AER and ANA), and iJO1366 (AER and ANA). Concurrently, excessive ethanol production was achieved in strains iTO977 (AER and ANA) and Yeast 7.6 (ANA). Here, eight of the twelve mutant strain schemes obtained have been identified and predicted to provide an overproduction of the desired metabolites, where the RBI-T2 and RBI-T3 algorithms yielded three and five optimal mutant strains, respectively.

## DISCUSSION

This research showed new regulatory-metabolic network models that combined GRNs and metabolic networks to design the optimal mutant strains. The proposed model was termed the RBI algorithm, categorized into three types according to the initial values of its TFs, namely RBI-T1, RBI-T2, and RBI-T3. This algorithm was the first continuous-type integrating approach that comprehensively incorporated Boolean rules within the empirical GRNs and GPR rules in the regulatory-metabolic network model. The reliability theory that was applied in the RBI algorithms allowed for the visualization of the relationship between transcription factors that regulated a gene, represented as a combination of parallel and series structures. This relationship could be expressed as a mathematical formula to quantify the reliability or activity of the regulated gene. Besides, the types of interactions of TFs, inhibition or activation, were fully considered in this model. Therefore, the RBI algorithms were more comprehensive than the earlier algorithms in modelling biology systems, especially the integration model of gene regulatory networks and metabolic networks. To identify the performance of the RBI algorithms, simulations were conducted in predicting the production rate of biomass and particular metabolites in strains of *E. coli* and *S. cerevisiae*. Subsequently, the prediction results of RBI were compared to the previous algorithms.

The effectiveness of the RBI algorithms in predicting biomass fluxes and indole production rates exhibited considerable variation among the three types: RBI-T1, RBI-T2, and RBI-T3. RBI-T1 exhibited robust predictive capability for biomass fluxes, attaining the lowest RMSE (0.088) compared to all algorithms, alongside competitive PCC (0.871) and R-squared (0.647) values. The obtained validity results demonstrated reliable performance with slight overestimation, as evidenced by a low bias of 0.059. In predicting indole production rates, RBI-T1 exhibited competitive RMSE (0.025) and bias (0.025) values. In addition, its PCC of 0.499 indicated a moderate positive correlation with empirical data,

and the R-squared of 0.500 demonstrated a moderate ability to explain empirical data variance. The results indicate that RBI-T1 has commendable predictive accuracy. Also, its prediction could explain the variation of the empirical results. RBI-T2 exhibited inadequate performance in biomass flux predictions, yielding sub-optimal results across all metrics. The RMSE of 0.484, the PCC of −0.635, and the bias of 0.453 indicated low accuracy and inconsistency. Nevertheless, RBI-T2 demonstrated superior performance in predicting indole production rates, outperforming all other algorithms, including PROM, TRFBA, and TRIMER, with a PCC of 0.808 and an R-squared value of 0.500. This indicated a strong positive correlation and an enhanced capacity to account for variance in empirical data. The RMSE (0.023) and bias (0.023) were superior to those of TRFBA but marginally lower than PROM and TRIMER. RBI-T3 demonstrated moderate effectiveness in biomass flux predictions, surpassing RBI-T2 but falling short of PROM, TRFBA, and TRIMER. The RMSE of 0.247 indicated acceptable accuracy, while the PCC of 0.296 and R-squared value of 0.473 suggested a moderate positive correlation and a reasonable ability to explain variance, accompanied by a slight overestimation reflected in a bias of 0.187. In predicting indole production rates, RBI-T3 demonstrated comparable performance to other proposed algorithms. Its prediction accuracy, based on the RMSE, R-squared, and biar values, was 0.029, 0.497, and 0.029, respectively, which were not much different from RBI-T1 and RBI-T2. However, its PCC of −0.753 indicated an inconsistent correlation with empirical data. The results underscored RBI-T3's accuracy strengths while also indicating areas for improvement in correlation regarding indole predictions. Generally, RBI-T1 was the most accurate algorithm for predicting the production rate of biomass for *E. coli* based on this simulation. For the prediction of indole production rate, RBI-T1 and RBI-T2 were equally good algorithms, with RBI-T2 slightly more consistent with the empirical results. In conclusion, RBI-T2 was the most adaptable and competitive alternative in simulations of Indole production conducted.

The results in the mutant strain of *S. cerevisiae* conducted indicated that RBI-T1 dominantly outperformed other algorithms, exhibiting enhanced predictive accuracy and reliability. The model demonstrated a low RMSE of 6.444, marginally exceeding that of OptFlux, the highest PCC of 0.819, and an R-squared value of 0.667, which signified that it explained 66.7% of the data variability. Additionally, it exhibited the lowest bias at 2.853, reflecting minimal systematic error. RBI-T2 exhibited inferior performance across all metrics relative to RBI-T1, recording a higher RMSE of 14.365, a moderate PCC of 0.305, and an R-squared value of 0.520. Nonetheless, it outperformed baseline models such as PROM and TRFBA. The elevated bias of 6.417 indicated a rise in systematic errors. RBI-T3 exhibited moderate performance, achieving an RMSE of 6.784, a robust PCC of 0.811, and an R-squared value of 0.657. This demonstrated an improvement over RBI-T2, although it fell short compared to RBI-T1. The bias of 3.254 was lower than that of RBI-T2 but higher than RBI-T1, indicating a moderate systematic error. RBI-T1 exhibited the best performance, followed by RBI-T3, and finally RBI-T2.

Based on the statements mentioned above, it was known that RBI-T1 outperformed RBI-T2 and RBI-T3 in predicting biomass production rate for *E. coli* and the particular metabolites in *S. cerevisiae*. Meanwhile, in the production rate of indole, the performance of RBI-T1 was quite competitive with RBI-T2 and RBI-T3, as noticed from the dominant validity results that are not much different from them. Meanwhile, RBI-T2 had a significantly lower performance compared to RBI-T1 and RBI-T3 in predicting the production rate of biomass and the particular metabolites in *S. cerevisiae*. However, in the indole prediction results, RBI-T2 showed a good performance and slightly better than RBI-T1. Finally, RBI-T3 showed good performance in biomass prediction and the mutant strains of *s. cerevisiae*. In those cases, the RBI-T3 algorithm was better than RBI-T2, although it was a slightly lower performance than RBI-T1. Nevertheless, the performance of RBIT3 required improvement in the Indole production, especially in correlation with the empirical results. In general, in the simulations that have been carried out, RBI-T1 had the best performance, followed by RBI-T3, and finally RBI-T2.

Moreover, based on the time complexity, all types of RBI algorithms had identical time complexity, namely $O(n^2)$, which was concluded as the quadratic time complexity *Younes et al. (2022)*. The process that caused the high time complexity in the RBI algorithms only occurred once during the RBI process, namely in determining the reference fluxes using the flux variability analysis. In addition, based on the simulations conducted to identify the running time required by the RBI algorithm, it was concluded that increasing the strain complexity did not significantly increase the running time requirements of this algorithm. Therefore, the RBI algorithm has quite good efficiency capabilities. This work has also utilized RBI algorithms to design optimum mutant strains of the microorganisms *E. coli* and *S. cerevisiae* in *in silico*. The simulation results indicated that those proposed algorithms suggested some TF knockout schemes capable of gaining the overproduction of ethanol and succinate. Out of the twelve schemes obtained, eight optimal mutant strain schemes outperformed the wild-type in producing the desired metabolite. In addition, some mutants exhibited elevated metabolite production relative to PROM and TRFBA. This implementation indicates that the suggested method is able to design the optimal mutant strains.

The proposed RBI algorithms can be applied to model the integration of gene regulatory networks and metabolic networks of microbes, plants, and even humans, which have huge complexity. This algorithm is the first to integrate the empirical GRNs into metabolic networks comprehensively. In addition, the empirical GRNs have a higher level of confidence compared to the inferred GRNs (*Larsen et al., 2019*; *Parise et al., 2021*). However, the availability of empirical GRNs is not as high as that of the inferred GRNs because the empirical GRNs resulted from direct observation, such as RNA-seq technologies (*Van Der Sande, Frölich & Van Heeringen, 2023*). Meanwhile, the inferred GRNs were deduced based on the gene expression data, which is abundantly available. Therefore, when datasets of the empirical GRNs are available, the regulatory-metabolic network model of microbes, plants, and even humans for designing the optimal mutant strains can be conducted using RBI algorithms.

Finally, the proposed RBI algorithms (RBI-T1, RBI-T2, RBI-T3) still have shortcomings. They do not involve the signaling networks in designing the mutant strains. This is because the signaling networks have no comprehensive datasets. In addition, the dynamic models constructed for signaling networks require parameters that are often not available (*Raman, 2021*). Nevertheless, the signaling networks play a role in the responses of microbes to environmental cues, stress conditions, and substrate availability (*Ananda, Daud & Zainudin, 2024*). Additionally, dynamic models are important in the metabolic process, as they can analyze the essential interactions and processes, such as the feedback inhibition of an enzyme or the signaling cascade activated in response to glucose ingestion (*Nilsson et al., 2022*). Therefore, attempting to incorporate signaling networks into the regulatory-metabolic network models has the potential to improve the prediction results in *in silico* metabolic engineering. Moreover, the omics dataset can be leveraged to obtain the parameters required by the RBI algorithms, for instance, obtaining the parameter needed from the gene expression data using biclusters approaches (*Huang et al., 2012, 2019*; *Sun & Huang, 2023*). Accordingly, the future improvement of the RBI algorithm can be conducted by including the signaling networks alongside GRNs and metabolic networks in *in silico* metabolic engineering. By involving the signaling networks, the metabolic system will be modeled more comprehensively. As a result, the prediction results obtained will be more accurate and robust.

## CONCLUSION

In this work, reliability-based integrating algorithms have been proposed for integrating the gene regulatory network and metabolic networks. This integration aims to construct a comprehensive model by involving the whole system of the GRNs and GPRs rules constructed in the Boolean equations. In addition, this algorithm was proposed as an attempt to overcome the limitations of the existing algorithms in handling Boolean equations. There are three types of the RBI algorithm, namely RBI-T1, RBI-T2, and RBI-T3, where the main difference among them is in determining the initial probability value of the transcriptional factors. In order to identify its performance, the RBI algorithm was validated based on accuracy and correlation to the actual results, as well as time complexity. Also, the RBI algorithm was compared to the state-of-the-art algorithm in predicting the production rate of biomass, indole, succinate, 2,3-butanediol, and ethanol. Based on those activities, all types of the RBI algorithm generally performed well, where RBI-T3 is the first-best algorithm, followed by RBI-T1 and RBI-T2. Accordingly, those algorithms can be suggested for designing the optimal mutant strains in *in silico* metabolic engineering. Based on the aforementioned statements, it could be concluded that the proposed RBI algorithms succeed in modelling the empirical GRNs and metabolic networks comprehensively for *in silico* metabolic engineering. Nevertheless, this algorithm still has limitations. All types of RBI do not involve signalling networks when designing the mutant strains. Therefore, this limitation can be used in future work to enhance the RBI algorithms.

### Funding

This work was supported by the Fundamental Research Grant Scheme (FRGS), grant number: FRGS/1/2021/ICT02/UKM/02/2 funded by the Ministry of Higher Education (MOHE), Malaysia. It was also supported by Geran Galakan Penyelidik Muda (GPPM), grant number: GGPM-2021-039, from Universiti Kebangsaan Malaysia. The funders had no role in study design, data collection and analysis, decision to publish, or preparation of the manuscript.

### Grant Disclosures

The following grant information was disclosed by the authors:
Fundamental Research Grant Scheme (FRGS): FRGS/1/2021/ICT02/UKM/02/2.
Ministry of Higher Education (MOHE), Malaysia.
Geran Galakan Penyelidik Muda (GPPM), Universiti Kebangsaan Malaysia:
GGPM-2021-039.

### Competing Interests

The authors declare that they have no competing interests.

### Author Contributions

- Ridho Ananda conceived and designed the experiments, performed the experiments, analyzed the data, performed the computation work, prepared figures and/or tables, and approved the final draft.
- Kauthar Mohd Daud performed the experiments, analyzed the data, authored or reviewed drafts of the article, and approved the final draft.
- Suhaila Zainudin performed the experiments, analyzed the data, authored or reviewed drafts of the article, and approved the final draft.

### Data Availability

The code is available in the Supplemental File.

### Supplemental Information

Supplemental information for this article can be found online at http://dx.doi.org/10.7717/peerj-cs.2880#supplemental-information.

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
