# Peer review of "RBI: a novel algorithm for regulatory-metabolic network model in designing the optimal mutant strain"

_PeerJ Computer Science, doi:10.7717/peerj-cs.2880_

## Round 0.1 · original submission · Major Revisions

The authors must work and improve the paper. Both reviewers have provided extensive critical comments which must be addressed in detail

Reviewer 1 ·

Basic reporting

Basic Reporting: The language throughout the manuscript is difficult to follow and is in need of substantial copyediting to be up to professional standards. I have made suggestions where it seems appropriate and where mistakes meaningfully affect clarity even to someone familiar with the field, but I am not attempting to provide comprehensive edits. Although I think, upon multiple read throughs, that I grasp the underlying motivation for using reliability theory to tackle this topic, the manuscript does not present a very coherent narrative that would make one inclined to think this is a good approach. To summarize, substantial revision of the writing by a copyeditor and a restructuring of the paper (primarily the introduction and the section on reliability theory) would be needed to make the paper acceptable for publication.

Minor comments:

Line 13: This suggests that Flux Balance Analysis was developed to accomplish a bioengineering goal, but this is misleading. A reader unfamiliar with FBA might come away thinking that (1) FBA is a wet lab technique, not a computational technique, and that (2) this is the only use of FBA. I would rephrase as, “Flux Balance Analysis (FBA) is a computational modeling framework that can be used to carry out in silico experiments for the optimization of metabolite overproduction from mutant strains,” or something along those lines.
Line 15: Make it clear that you are referring to previous work, not your own.
Line 17: Meaning of “whereas” and following clause a bit unclear to me. Perhaps, “These algorithms heavily depend on gene expression data (GED), but their effectiveness is reduced by inconsistencies between GED and GRNs and incorrect genes-proteins-reactions (GPR) rules.”
Line 20 (and elsewhere): The author’s use of “involve” is not idiomatic. Perhaps “integrate” would be better here?
Line 48: “Intensitivity,” not “insensitive.”
Line 49: This sentence appears incomplete. Please rewrite.
Line 76: I think this section is misleading, because it conflates metabolic modeling methods that leverage inferred GRNs (enumerated at the beginning of the paragraph) with the inference of GRNs itself. When you refer to “those algorithms,” it can only be read as referring to methods like CoRegFlux, IDREAM, etc., but the citations provided at the end of the paragraph are specifically about inferred GRNs, not their use in predicting metabolic. Please rewrite this section.
Line 83: “Resulted comprehensively” needs to be rewritten.
Line 119: A brief what? Also, this is formatted as a separate section, but is this not part of the introduction?
Line 163: Meaning of “effortless” here is unclear. Perhaps “essential”?
Line 664 – 668: This reference is duplicated.

Experimental design

Experimental Design: Although the methods are described in detail, a close-examination of the comparison the authors made between their results and those generated by competing algorithms shows that there are a number of serious issues. Moreover, underlying data from an inappropriate, non-scholarly source is used and there are basic statistical misunderstandings in the analysis. Proper sourcing of input data and more thorough and justifiable comparisons with competing algorithms would be needed to make the paper suitable for publication.

The use of a gene expression dataset of unknown providence, deposited on Kaggle (first mention: line 206), is inappropriate. I do not see any obvious indication of where the “publicly available” data used to make this dataset originally comes from. Did it come from publications? If so, these need to be cited. What are the SRR numbers or equivalent identifiers? What are the details of the post-processing of the raw expression data? This does not meet publication standards. If the authors want to use this dataset to validate the use of their method in yeast, then we need references, identifiers, and analysis methods in the manuscript.

Table 4: Multiple versions of TRIMER are presented in the cited publication from which the experimental data are taken, but the authors have chosen to report the results of the worst performing one (Trimer-C), which I think is inappropriate and misleading. Optimal performance is seen with the TRIMER-B sFBA combination, which has a PCC of 0.906 (much higher than the 0.537 in Table 2 and also higher than the 0.755-0.820 range for the RBI methods).

Tables 4 & 6: The authors appear to be using this p-value calculation using a t-test as a way of assessing whether the distribution of values from a particular method is statistically significantly likely to be sampled from the same distribution as the actual measured values, but this is not valid. Put simply, when testing a null hypothesis, a value over one’s chosen alpha (in this case 0.05, I’m assuming) is NOT evidence FOR the null hypothesis, which is how it is being used here. Please remove this analysis from the paper as it does not tell the reader anything about the reliability of these different algorithms.
Table 6: The “Actual” column does not refer to empirically measured values – rather, it refers to the predicted max/min fluxes possible given an assumed 99% or 50% optimal biomass accumulation by FVA in an empirically validated model. First off all, citations to the sources of these original models and experiments should be provided. Second, the authors should be clearer about what this “validation” demonstrates. It demonstrates that their method may outperform existing methods in predicting the behavior of an experimentally validated model of metabolism. It doesn’t show that the RBI method actually does a better job of predicting experimental data directly – for that, the authors should look at the original experiments and try to compare model outputs with these. Because of this, I am not inclined to interpret the positive results for RBI-T1 in this table as compelling proof that the algorithm is actually extracting signal more effectively than the others.

Validity of the findings

Validity of the findings: Overall, the claim being made by the authors is that the presented family of algorithms is superior to existing methods. The issues I highlight in the experimental design section make me strongly question whether this is the case and also whether the competing algorithms are being fairly assessed, so I do not think this claim is supported and I cannot in good faith recommend this manuscript for publication.

Code to reproduce this study’s results is not provided. Even if the description of the methods provided in the paper were adequately detailed, a computational methods paper that does not share the underlying code is not appropriate.

Additional comments

Summary: In this paper, the authors present several algorithms (RBI-T1 through T3) based on reliability theory for integrating information from gene-regulatory networks and gene-expression data into metabolic flux predictions. Their algorithms are benchmarked by comparison with some preexisting empirical data as well as the predictions of other methods (e.g. TRIMER and TRFBA).

Overall assessment: Although the conceptual basis for the proposed method has promise, the comparison of the RBI-T1 through T3 algorithms with existing methods and empirical data is hampered by a number of critical flaws and ultimately fails to support the author’s argument that they have developed a method that “generally outperformed” existing ones. The comparison of production fluxes in Table 6 is not, I think, a meaningful validation because the values being compared to are themselves computational predictions, not the actual (and available, I might add) empirical values. This leaves us with the results in Table 4; however, digging into this table reveals that one of the five evaluated metric (p-value of a t-test) is not statistically meaningful and the results of at least one of the competing algorithms (TRIMER) are misrepresented in a way that makes it look worse performing than it actually is. The evidence for this method’s superiority to existing algorithms is, therefore, rather lacking in the present manuscript. Code and critical implementation details are also lacking, making the findings unreproducible. For these reasons, I must recommend rejection of the manuscript in its present form.

Reviewer 2 ·

Basic reporting

The manuscript introduces a novel algorithm, RBI, which integrates gene regulatory networks (GRNs) and metabolic networks, a significant contribution to in silico metabolic engineering. The authors consider both non-inference GRNs and inference GRNs, acknowledging the high confidence associated with non-inference GRNs derived from direct experimental data. The manuscript provides a thorough validation of the RBI algorithm based on accuracy, robustness, and time complexity, which strengthens the credibility of the proposed method. However, the manuscript does not incorporate signaling networks in the design of mutant strains, which could enhance the model's comprehensiveness and accuracy. Therefore, future work could benefit from including signaling networks alongside GRNs and metabolic networks to improve prediction results in in silico metabolic engineering. I think this article is of interest to readers in the field. However, there are some comments listed below might improve the manuscript.

Experimental design

1. The authors validate the performance of the RBI algorithm by comparing it with existing algorithms. It is recommended that the authors provide more details about the validation process, including the datasets used, parameter settings, and the basis for the selection of the validation metrics.
2. The RBI algorithm requires optimisation of the parameters α, β, γ. The authors are requested to elaborate on the biological significance of these parameters and how the optimal values of these parameters can be determined by meta-heuristic optimisation methods.
3. The article states that the time complexity of the RBI algorithm is O(n^2). The authors are requested to provide more discussion on the efficiency of the algorithm, especially its performance when dealing with large-scale gene regulatory networks and metabolic networks.
4. The article uses specific microbial models for simulations. Can the authors discuss how representative these models and datasets are and whether the RBI algorithm can generalise to other types of microbes and metabolic networks?

Validity of the findings

No comment

Additional comments

1. The abstract could be made more concise by directly stating the main outcomes and the significance of the RBI algorithm without excessive detail.
2. The introduction could benefit from a more detailed background on the current challenges in in silico metabolic engineering, setting the stage for the importance of the RBI algorithm.
3. The English language should be improved to ensure that an international audience can clearly understand your text. Some examples where the language could be improved. For instance, "The demand for bacterial cellulose production has increased significantly." could be improved for clarity to "There has been a significant increase in the demand for bacterial cellulose production."; "Consequently, those algorithms may perform poorly." could be rephrased to "As a result, the performance of these algorithms may be suboptimal."

---

## Round 0.2 · Major Revisions

The authors must do a further major revision to address the remaining comments

Reviewer 1 ·

Basic reporting

Fine, but with the following comments:

I will not go through the grammar of the introduction line-by-line, but it should be revisited again to make it more idiomatic and readable. The abstract as well. The Results, Methods, and Discussion, on the other hand, read well enough.

“Inference” vs. “non-inference” GRN do not appear to be terms used in the literature, though I understand the meaning the authors are going for. If this is a novel categorization that the authors have come up with, please make this clear. If not, please use names that are common in the field. Also, it should be “inferred” or “non-inferred,” not “inference” or “non-inference.” Also, the distinction is between inference and explicit measurement, so perhaps “inferred” and “empirical” would be a better distinction.

Line 42: Please cite early work describing and developing FBA, not just recent reviews on the topic.

Experimental design

Much better described and justified than in the previous revision. However:

The authors have shared their code, but it would be helpful to include ready-to-run examples in the code. Simply initializing the MATLAB scripts, as recommended in the attached README, was not sufficient, and after some guessing at renaming files, I was unable to get things to run. Please include either detailed instructions on how to reproduce the results reported in the paper or have the examples set up to run to begin with.

Validity of the findings

Following comments need to be addressed:

It is unclear to me what datasets are being used for the metaheuristic optimization of the S. cerevisiae and E. coli RBI-T1-3 models. Is data from, say, the E. coli TF-knockouts (used for validation, Table 4) included in this initial optimization? If so, this would not constitute a real validation. The same applies for the other validations that were performed. Please clarify in the text, and if there is overlap presently, please rerun validations such that the model has not “seen” anything related to the validation before it is done.

I am a little puzzled by the results presented in Table 5 for TRFBA specifically. Is there an intuitive reason why TRFBA always makes the same prediction?

Additional comments

Summary: In this revised manuscript, the authors present the formulation and validation of their reliability based method for integrating gene regulatory network information with metabolic network information, and then using these integrated models to predict fluxes. I previously recommended rejection of this manuscript, but believe that substantial improvements have been made in the description, validation, and reporting of the results. I still have some concerns with the manuscript, which I have detailed in my review.

Reviewer 2 ·

Basic reporting

I thank the authors for their addressing my comments.

Experimental design

no comment

Validity of the findings

no comment

Additional comments

no comment

---

## Round 0.3 · Minor Revisions

There are some remaining minor concerns that need to be addressed.

Reviewer 1 ·

Basic reporting

Language has been improved since the last revision and I believe is clear enough for publication. However, ready-to-run test cases are still not provided. The revision I'm requesting is the same as last time - please include explicit instructions, together with correctly named model files, on how to run a test case of the algorithm. I tried renaming various included files to try to match the requirements of the code, but with no instructions, it is still too opaque for publication and sharing.

Experimental design

No comment

Validity of the findings

No comment

---

## Round 0.4 · accepted · Accept

The authors have addressed all the remaining concerns and I recommend accepting this manuscript.

Reviewer 1 ·

Basic reporting

Code can now be run relatively easily, so my only problem has been fixed. Publishable.

Experimental design

No change from last time - publishable.

Validity of the findings

No change from last time - publishable.

Additional comments

I thank the authors for their clearer instructions as regards the code. I think the manuscript is suitable for publication. I do recommend, though I do not think this warrants a "revision," that the final typeset version of the publication have the attached code shared as a GitHub repository, since this is standard practice in the field.